# MIDI-DDSP: DETAILED CONTROL OF MUSICAL PERFORMANCE VIA HIERARCHICAL MODELING

**Yusong Wu**[1], **Ethan Manilow**[2,4], **Yi Deng**[3], **Rigel Swavely**[4], **Kyle Kastner**[1],
**Tim Cooijmans**[1], **Aaron Courville**[1], **Cheng-Zhi Anna Huang**[*1,4] , **Jesse Engel**[*4]
* Equal Contribution
[1]Mila, Quebec Artificial Intelligence Institute, Université de Montréal,
[2]Northwestern University, [3]New York University, [4]Google Brain
`wu.yusong@mila.quebec`
`{emanilow, annahuang, jesseengel}@google.com`

## ABSTRACT

Musical expression requires control of both *what* notes are played, and *how* they are performed. Conventional audio synthesizers provide detailed expressive controls, but at the cost of realism. Black-box neural audio synthesis and concatenative samplers can produce realistic audio, but have few mechanisms for control. In this work, we introduce MIDI-DDSP a hierarchical model of musical instruments that enables both realistic neural audio synthesis and detailed user control. Starting from interpretable Differentiable Digital Signal Processing (DDSP) synthesis parameters, we infer musical notes and high-level properties of their expressive performance (such as timbre, vibrato, dynamics, and articulation). This creates a 3-level hierarchy (notes, performance, synthesis) that affords individuals the option to intervene at each level, or utilize trained priors (performance given notes, synthesis given performance) for creative assistance. Through quantitative experiments and listening tests, we demonstrate that this hierarchy can reconstruct high-fidelity audio, accurately predict performance attributes for a note sequence, independently manipulate the attributes of a given performance, and as a complete system, generate realistic audio from a novel note sequence. By utilizing an interpretable hierarchy, with multiple levels of granularity, MIDI-DDSP opens the door to assistive tools to empower individuals across a diverse range of musical experience. [1]

## 1 INTRODUCTION

Generative models are most useful to creators if they can generate realistic outputs, afford many avenues for control, and easily fit into existing creative workflows (Huang et al., 2020). Deep generative models are expressive function approximators, capable of generating realistic samples in many domains (Ramesh et al., 2021; Brown et al., 2020; van den Oord et al., 2016), but often at the cost of interactivity, restricting users to rigid black-box input-output pairings without interpretable access to the internals of the network. In contrast, structured models chain several stages of interpretable intermediate representations with expressive networks, while still allowing users to interact throughout the hierarchy. For example, these techniques have been especially effective in computer vision and speech, where systems are optimized for both realism and control (Lee et al., 2021b; Chan et al., 2019; Zhang et al., 2019; Wang et al., 2018a; Morrison et al., 2020; Ren et al., 2020).

For music generation, despite recent progress, current tools still fall short of this ideal (Figure 1, right). Deep networks can either generate realistic full-band audio (Dhariwal et al., 2020) or provide detailed controls of attributes such as pitch, dynamics, and timbre (Défossez et al., 2018; Engel et al.,

---

[1]Online resources:
Code: `https://github.com/magenta/midi-ddsp`
Audio Examples: `https://midi-ddsp.github.io/`
Colab Demo: `https://colab.research.google.com/github/magenta/midi-ddsp/blob/main/midi_ddsp/colab/MIDI_DDSP_Demo.ipynb`

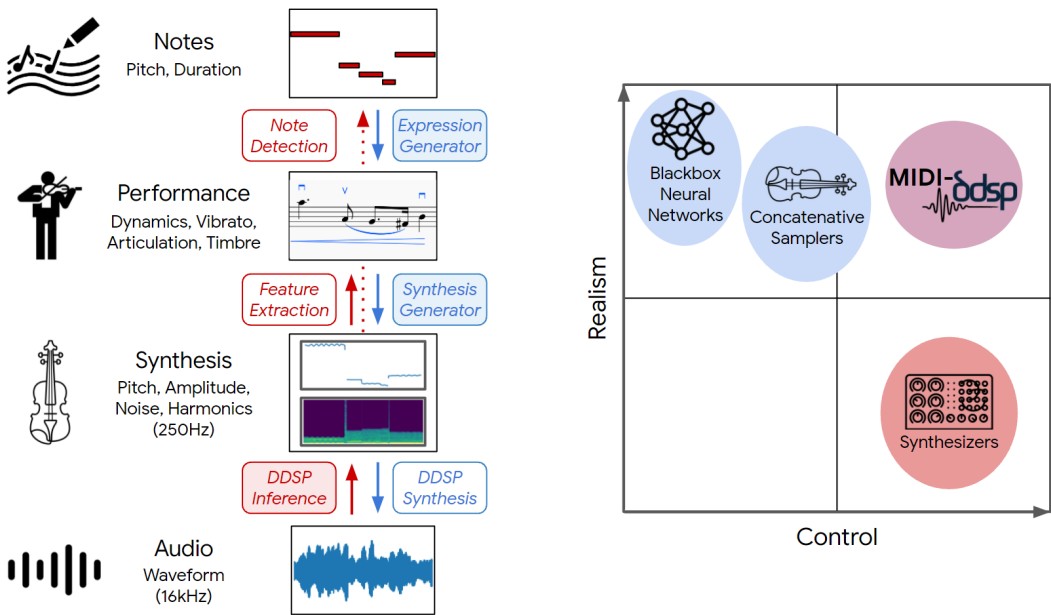

Figure 1: (Left) The MIDI-DDSP architecture. MIDI-DDSP extracts interpretable features at the performance and synthesis levels, building a modeling hierarchy by learning feature generation at each level. Red and blue components indicate encoding and decoding respectively. Shaded boxes represent modules with learned parameters. Both expression features and notes are extracted directly from synthesis parameters. (Right) Synthesizers have wide range of control, but struggle to convey realism. Neural audio synthesis and concatenative samplers can produce realistic audio, but have limited control. MIDI-DDSP enables both realistic neural audio synthesis and detailed user control.

2019; 2020a; Hawthorne et al., 2019; Wang & Yang, 2019) but not both. Many existing workflows use the MIDI specification (Association et al., 1996) to Conventional DSP synthesizers (Chowning, 1973; Roads, 1988) provide extensive control but make it difficult to generate realistic instrument timbre, while concatenative samplers (Schwarz, 2007) play back high-fidelity recordings of isolated musical notes, but require manually stitching together performances with limited control over expression and continuity.

In this paper, we propose MIDI-DDSP, a hierarchical generative model of musical performance to provide both realism and control (Figure 1, left). Similar to conventional synthesizers and samplers that use the MIDI standard (Association et al., 1996), MIDI-DDSP converts note timing, pitch, and expression information into fine-grained parameter control of DDSP synthesizer modules.

We take inspiration from the hierarchical structure underlying the process of creating music. A composer writes a piece as a series of notes. A performer interprets these notes through a myriad of nuanced, sub-second choices about articulation, dynamics, and expression. These expressive gestures are realized as audio through the short-time pitch and timbre changes of the physical vibration of the instrument. MIDI-DDSP is built on a similar 3-level hierarchy (notes, performance, synthesis) with interpretable representations at each level.

While the efficient DDSP synthesis representation (low-level) allows for high-fidelity audio synthesis (Engel et al., 2020a), users can also control the notes to be played (high-level), and the expression with which they are performed (mid-level). A qualitative example of this is shown in Figure 2, where a given performance on violin is manipulated at all three levels (notes, expression, synthesis parameters) to create a new realistic yet personalized performance.

As seen in Figure 1 (left), MIDI-DDSP can be viewed similarly to a multi-level autoencoder. The hierarchy has three separately trainable modules (*DDSP Inference*, *Synthesis Generator*, *Expression Generator*) and three fixed functions/heuristics (*DDSP Synthesis*, *Feature Extraction*, *Note Detection*). These modules enable MIDI-DDSP to conditionally generate at any level of the hierarchy,

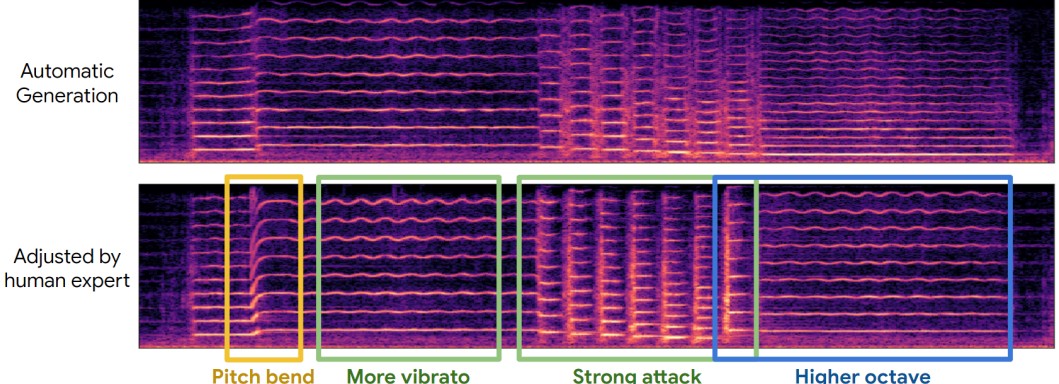

Figure 2: An example of detailed user control. Given an initial generation from the full MIDI-DDSP model (top), an expert musician can adjust notes (blue), performance attributes (green), and low-level synthesis parameters (yellow) to craft a personalized expression of a musical piece (bottom).

providing creative assistance by filling in the details of a performance, synthesizing audio for new note sequences, or even fully automating music generation when paired with a separate note generating model.

It is important to note that the system relies on pitch detection and note detection, so is currently limited to training on recordings of single monophonic instruments. This approach has the potential to be extend to polyphonic recordings via multi-instrument transcription (Hawthorne et al., 2021; Engel et al., 2020b; Bittner et al., 2017) and multi-pitch tracking, which is an exciting avenue to explore for future work. Finally, we also show that each stage can be made conditional on instrument identity, training on 13 separate instruments with a single model.

For clarity, we summarize the core contributions of this work:

- We propose MIDI-DDSP, a 3-level hierarchical generative model of music (notes, performance, synthesis), and train a single model capable of realistic audio synthesis for 13 different instruments. (Section 3)
- *Expression Attributes:* We introduce heuristics to extract mid-level per-note expression attributes from low-level synthesis parameters. (Figure 4)
- *User Control:* Quantitative studies confirm that manipulating the expression attributes creates a corresponding effect in the synthesizer parameters, and we qualitatively demonstrate the detailed control that is available to users manipulating all three levels of the hierarchy. (Table 2 and Figure 2)
- *Assistive Generation:* Reconstruction experiments show that MIDI-DDSP can make assistive predictions at each level of the hierarchy, accurately resynthesizing audio, predicting synthesis parameters from note-wise expression attributes, and auto-regressively predicting note-wise expression attributes from a note sequence. (Tables 1a, 1b, 1c)
- *Realistic Note Synthesis:* An extensive listening study finds that MIDI-DDSP can synthesize audio from new note sequences (not seen during training) with higher realism than both comparable neural approaches and professional concatenative sampler software. (Figure 5)
- *Automatic Music Generation:* We demonstrate that pairing MIDI-DDSP with a pretrained note generation model enables full-stack automatic music generation. As an example, we use Coconet (Huang et al., 2017) to generate and synthesize novel 4-part Bach chorales for a variety of instruments. (Figure 6)

## 2 RELATED WORK

**Note Synthesis**. Existing neural synthesis models allow either high-level manipulation of note pitch, velocity, and timing (Hawthorne et al., 2019; Kim et al., 2019; Wang & Yang, 2019; Manzelli et al.,

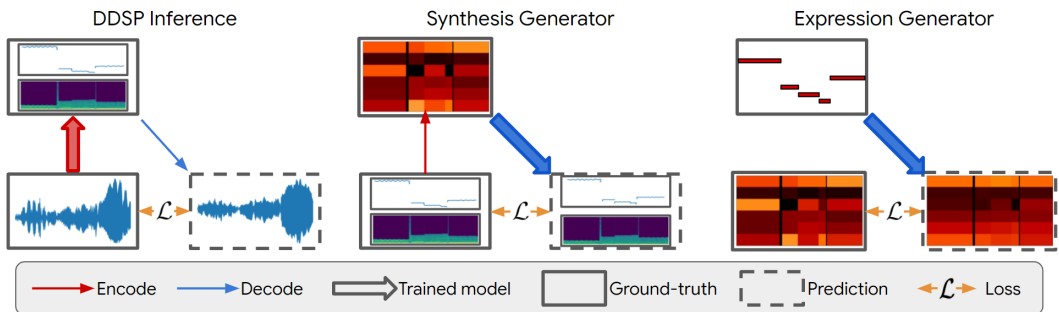

Figure 3: Separate training procedures for the three modules in MIDI-DDSP. (Left) The DDSP Inference module predicts synthesis parameters from audio and is trained via an audio reconstruction loss on the resynthesized audio. (Middle) The Synthesis Generator module predicts synthesis parameters from notes and their expression attributes (shown as a 6-dimensional color map) and is trained via a reconstruction loss and an adversarial loss. (Right) The Expression Generator module autoregressively predicts note expression given a note sequence and is trained with teacher forcing. Encoding processes are shown in red, and decoding processes are shown in blue and loss calculations are shown in yellow. Thicker arrows indicate the process that is being trained in each level. Ground-truth data are shown in solid frames, while model predictions are shown in dashed frames.

2018), or low-level synthesis parameters (Jonason et al., 2020; Castellon et al., 2020; Blaauw & Bonada, 2017). MIDI-DDSP connects these two approaches by enabling both high-level note controls and low-level synthesis manipulation in a single system.

Most related to this work is MIDI2Params (Castellon et al., 2020), a hierarchical model that autoregressively predicts frame-wise pitch and loudness contours to drive the original DDSP autoencoder (Engel et al., 2020a). MIDI-DDSP builds on this work by adding an additional level of hierarchy for the note expression, training a new more accurate DDSP base model, and explicitly modeling the synthesizer coefficients output by that model, rather than the pitch and loudness inputs to the model. We extensively compare to our reimplementation of MIDI2Params as a baseline throughout the paper.

**Hierarchical Audio Modelling**. Audio waveforms have dependencies over timescales spanning several orders of magnitude, lending themselves to hierarchical modeling. For example, Dieleman et al. (2018) and Dhariwal et al. (2020) both choose to encode audio as discrete latent codes at different time resolutions, and apply autoregressive models as priors over those codes. MIDI-DDSP applies a similar approach in spirit, but constructs a hierarchy based on semantic musical structure (note, performance, synthesis), allowing interpretable manipulation by users.

**Expressive Performance Analysis and Synthesis**. Many prior systems pair analysis and synthesis functions to capture expressive performance characteristics (Canazza et al., 2004; Yang et al., 2016; Shih et al., 2017). Such methods often use heuristic functions to generate parameters for driving synthesizers or selecting and modifying sample units. MIDI-DDSP similarly uses feature extraction, but each level is paired with a differentiable neural network function that directly learns the mapping to expression and synthesis controls for more realistic audio synthesis.

## 3 MODEL ARCHITECTURE

### 3.1 DDSP SYNTHESIS AND INFERENCE

Differentiable Digital Signal Processing (DDSP) (Engel et al., 2020a) enables differentiable audio synthesis by using a harmonic plus noise model (Serra & Smith, 1990). Full details are provided in Appendix B.1. Briefly, an oscillator bank synthesizes a harmonic signal from a fundamental frequency $f_0(t)$, a base amplitude $a(t)$, and a distribution over harmonic amplitudes $\boldsymbol{h}(t)$, where the dimensionality of $\boldsymbol{h}$ is the number of harmonics. The noise signal is generated by filtering uniform noise with linearly spaced filter banks, where $\boldsymbol{\eta}(t)$ represents the magnitude of noise output

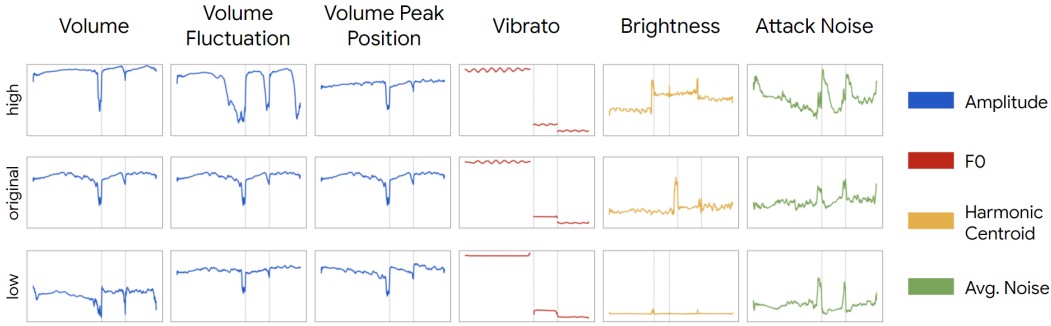

Figure 4: In MIDI-DDSP, manipulating note-level expression can effectively change the synthesis-level quantities. We show by taking a test-set sample (middle row) and adjusting each expression control value to lowest (bottom row) and highest (upper row), how each synthesis quantities (rightmost legend) would change. The dashed gray line in each plot indicates the note boundary.

from each filter in time. In this study, we use 60 harmonics and 65 noise filter banks, giving 127 total synthesis parameters each time frame ($s(t) = (f_0(t), a(t), h(t), \eta(t))$). The final audio is the addition of harmonic and noise signals.

Since the synthesis process is differentiable, Engel et al. (2020a) demonstrate that it is possible to train a neural network to predict the other synthesis parameters given $f_0(t)$ and the loudness of the audio, and optimize a multi-scale spectral loss (Wang et al., 2019; Engel et al., 2020a) of the resynthesized audio (Figure 3 left). $f_0(t)$ is extracted by a pre-trained CREPE model (Kim et al., 2018), and the loudness is extracted via an A-weighting of the power spectrum (Hantrakul et al., 2019; McCurdy, 1936).

We extend this work for our DDSP Inference module, by providing an additional input features extracted by a CNN (Lecun et al., 1998) from log-scale Mel-spectrogram of the audio, that produces higher quality resynthesis (Table 1a). Full architectural details are provided in Appendix B.2.

## 3.2 EXPRESSION CONTROLS

We aim to model aspects of expressive performance with a continuous variable. For example, this enables a performer to choose *how loud* the note should be performed, or *how much* vibrato to apply. We define a 6-dimensional vector, $e_i$, for each note, $n_i$, where each dimension corresponds to one of the six expression controls (detailed in Appendix B.3), scaled within $[0, 1]$. These are extracted from synthesis parameters $s(t)$ and applied within the $i$th note, $n_i$, in a note sequence:

**Volume**: Controls the volume of a note, extracted by taking average amplitude over a note.

**Volume fluctuation**: Determines the magnitude of a volume change across a note. Used with the volume peak position described below, this can make a note crescendo or decrescendo. This is extracted by calculating the standard deviation of the amplitude over a note.

**Volume peak position**: Controls where, over the duration of a note, the peak volume occurs. Zero value corresponds to decrescendo notes, whereas one corresponds to crescendo notes. The volume peak position is extracted by calculating the relative position of maximum amplitude in the note.

**Vibrato**: Controls the extent of the vibrato of a note. Vibrato is a musical technique defined by pulsating the pitch of a note. Vibrato is extracted by applying Discrete Fourier Transform (DFT) on the fundamental frequency $f_0(t)$ in a note and selecting the peak amplitude.

**Brightness**: Controls the timbre of a note where higher values correspond to louder high-frequency harmonics. Brightness is determined by calculating the average harmonic centroid of a note.

**Attack Noise**: Controls how much noise occurs at the start of the note (the attack), e.g., the fluctuation of string and bow. Attack noise can determine whether two notes sound consecutively or separately. Attack noise is extracted by taking a note's average noise magnitude in the first ten frames (40ms).

### 3.3 SYNTHESIS GENERATOR

Given the output of the per-note Expression Controls, $e_i$ for $i = 1, ..., I$ notes, and a corresponding note sequence, $n_i$, the Synthesis Generator predicts the frame-level synthesis parameters that, in turn, generate audio. Note expression controls are unpooled (repeated) over the duration of the corresponding note to make a conditioning sequence, $c(t) = [(e_1, n_1), ..., (e_I, n_I)]$, with the same length as the fundamental frequency curve, $f_0(t)$.

The Synthesis Generator, $g_\theta$, is an autoregressive recurrent neural net (RNN) is used to predict a fundamental frequency, $\hat{f}_0(t)$ given conditioning sequence, and a convolutional generative adversarial network (GAN), $g_\phi$, is used to predict the other synthesis parameters given conditioning sequence and generated fundamental frequency:

$$\hat{f}_0(t) = g_\theta(c(t)), \quad \hat{a}(t), \hat{h}(t), \hat{\eta}(t) = g_\phi(c(t), \hat{f}_0(t)), \tag{1}$$

where $\theta$ denotes trainable parameters in the autoregressive RNN, and $\phi$ indicates trainable parameters in the GAN. Architectural details for both of these details is provided in Appendix B.4. The autoregressive RNN is trained using cross-entropy loss $\mathcal{L}_{ce}$. The generator of the GAN is trained by a multi-scale spectral loss $\mathcal{L}_{spec}$ (Eq. 13-14) and an adversarial objective consisting of a least-squares GAN (LSGAN) $\mathcal{L}_{lsgan}$ (Mao et al., 2017) loss and a feature matching loss $\mathcal{L}_{fm}$ (Appendix B.4, Eqs. 15-18) (Kumar et al., 2019). Thus, the total loss applied to the Synthesis Generator can be written as:

$$\mathcal{L} = (\mathcal{L}_{ce} + \mathcal{L}_{spec}) + \alpha(\mathcal{L}_{lsgan} + \gamma\mathcal{L}_{fm}) \tag{2}$$

where $\alpha$ and $\gamma$ are settable hyperparameters that control the overall GAN loss and feature matching loss, respectively. For training, the ground-truth $f_0(t)$ is input to the GAN, thus there is no gradient from the GAN to the autoregressive RNN.

### 3.4 EXPRESSION GENERATOR

The Expression Generator uses an autoregressive RNN to predict note expression controls from note sequence (Appendix B.6). A single-layer bidirectional GRU extracts context information from input, and a two-layer autoregressive GRU generates note expression. The Expression Generator is trained by mean square error (MSE) loss between ground-truth note expression and teacher-forced prediction (Figure 3, right). The note sequence used to train the Expression Generator can either be extracted or comes from human labels. To show the full potential of MIDI-DDSP, we use the ground-truth note boundary label from dataset in all experiments for best accuracy. However, in future work note transcription models can be used to provide the note labels.

## 4 EXPERIMENTS

The structured hierarchy and explicit latent representations used in MIDI-DDSP benefit music control as well as music modeling. We design a set of experiments to answer the following questions: First, does the system generate realistic audio, and if so, how does each module contribute? How does this compare to existing systems? And, second, is the system capable of enabling note-level, performance-level, and synthesis-level control? How effective are these controls?

### 4.1 DATASET

To demonstrate modeling a variety of instruments, we use the URMP dataset (Li et al., 2018), a publicly-available audio dataset containing monophonic solo performances of a variety of instruments. URMP is widely used in music synthesis research (Bitton et al., 2020; Hayes et al., 2021; Zhao et al., 2019; Engel et al., 2020b). The URMP dataset contains solo performance recordings and ground truth note boundaries from 13 string and wind instruments, which allows us to test MIDI-DDSP on many different instruments. The recordings in the URMP dataset are played by students, and the performance quality is substantially lower compared to virtuoso datasets used in

Table 1: Each module in MIDI-DDSP produces a high-quality reconstruction and accurate prediction. We show reconstruction accuracy of each MIDI-DDSP module against a comparable method.

| Model | Spectral Loss | Model | RMSE | Models | RMSE |
|---|---|---|---|---|---|
| DDSP Inference | **4.28** | Synthesis Generator | **0.19** | Expression Generator | **0.14** |
| Engel et al. (2020a) | 5.00 | MIDI2Params | 0.26 | MIDI2Params | 0.23 |
| (a) | | (b) | | (c) | |

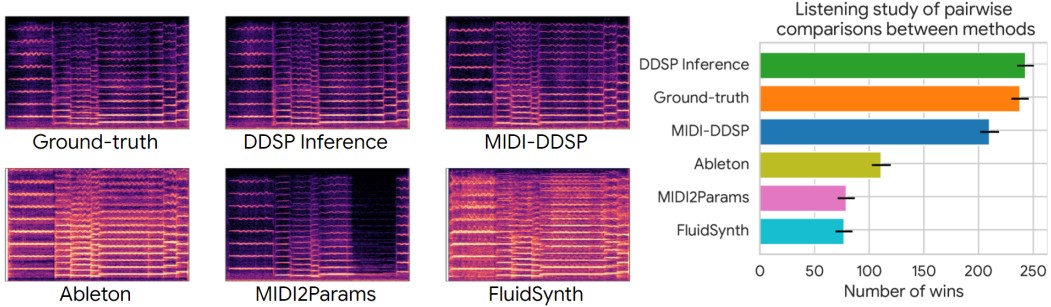

Figure 5: (left) Comparing the log-scale Mel spectrograms of synthesis results from test-set note sequences, MIDI-DDSP synthesizes more realistic audio (more similar to ground-truth and DDSP Inference) than prior score-to-audio work MIDI2Params (enlarged in Figure 7). The synthesis quality is also reflected in the listening study (right), where the MIDI-DDSP synthesis is perceived as more realistic than the professional concatenative sampler Ableton and the freely available FluidSynth.

other work (Hawthorne et al., 2019). The URMP dataset contains 3.75 hours of 117 unique solo recordings, where 85 recordings in 3 hours are used as the training set, and 35 recordings in 0.75 hours are used as the hold-out test set.

## 4.2 MODEL ACCURACY

Modules in MIDI-DDSP can accurately reconstruct output at multiple levels of the hierarchy (Figure 3). We evaluate the reconstruction quality of MIDI-DDSP by evaluating each module, and report the average value across all test set samples in Table 1.

**DDSP Inference** We measure the difference between reconstruction and ground-truth in the audio spectral loss for our DDSP Inference module and compared it with the original DDSP autoencoder. As shown in Table 1a, with an additional CNN to extract features, the DDSP Inference module can reconstruct audio more accurately than the original DDSP Autoencoder.

**Synthesis Generator** We predict synthesis parameters from ground-truth note expression and then extract note expression back from the generated synthesis parameters. We measure the root mean square error (RMSE) between note expressions. The prior approach MIDI2Params directly generates synthesis parameters from notes and does not have access to note expressions. However, we can extract note expressions from the generated synthesis parameters and compare them to ground truth. As shown in Table 1b, the Synthesis Generator can faithfully reconstruct the input note expression, whereas without access to note expression, MIDI2Params generates larger error.

**Expression Generator** We take ground-truth MIDI and evaluate the likelihood of the ground-truth note expressions under the model. As the Expression Generator is autoregressive, we use teacher-forcing to sequentially accumulate the squared error note by note. The total error thus computed can be interpreted as a log-likelihood. We again compare to MIDI2Params, where we autoregressively condition its own output within and on ground-truth across notes to obtain a note-wise metric. That is, MIDI2Params sees the ground truth of past notes, but sees its own output for the current note. As shown in Table 1c, the Expression Generator can accurately predict the note expression. In

Table 2: The note expression outputs are strongly correlated with input adjustment. The Pearson correlation $r$-values are shown in the table (all entries $p < 0.0001$). The bold numbers indicate a Pearson $r$-value larger than 0.7, which we consider to indicate strong correlation between the input control and the respective output quantity. For simplicity, only four instruments are shown. More results can be found in Table 7.

|  | Volume | Vol. Fluc. | Vol. Peak Pos. | Vibrato | Brightness | Attack Noise |
|---|---|---|---|---|---|---|
| All instruments | **.97** | **.78** | .57 | **.70** | **.92** | **.93** |
| Violin | **.99** | **.84** | **.80** | **.86** | **.96** | **.97** |
| Viola | **.98** | **.74** | **.70** | **.82** | **.98** | **.97** |
| Cello | **.97** | .64 | .54 | **.74** | **.98** | **.94** |
| Double bass | **.98** | **.85** | .34 | **.84** | **.99** | **.95** |

comparison, MIDI2Params without performance-level modeling suffers from predicting the note expression with a higher error when compared to our frame-wise sequence models.

### 4.3 Audio Quality Evaluation by Human Listeners

We evaluate the audio quality of MIDI-DDSP via a listening test. We compare ground truth audio from the URMP dataset to MIDI-DDSP and four other sources: a stripped down version of our system, containing just our DDSP Inference module (Section 3.1), MIDI2Params (Castellon et al., 2020), and two concatenative samplers: FluidSynth and Ableton (detailed in Appendix D.1). DDSP Inference infers synthesis parameters from the ground truth audio; it serves as an upper bound on what is attainable with MIDI-DDSP, which has to predict expression and synthesis parameters from MIDI. MIDI2Params is prior work that synthesizes audio from MIDI by predicting frame-wise loudness and pitch contour, which is fed as input to a DDSP autoencoder.

Participants in the listening test were presented with two 8-second clips, and asked which clip sounded more like a person playing on a real violin, on a 5-point Likert scale. We collected 960 ratings, with each source involved in 320 pair-wise comparisons. Figure 5 shows the number of comparisons in which each source was selected as more realistic. According to a post-hoc analysis using the Wilcoxon signed-rank test with Bonferroni correction (with $p < 0.01/15$), the orderings shown in Figure 5 (right) are all statistically significant with the exception of ground truth versus DDSP Inference and MIDI2Params versus FluidSynth (Table 6). In particular, MIDI-DDSP was significantly preferred over MIDI2Params, Ableton, and FluidSynth.

The difference among the sources can also be seen from visual inspection of the spectrograms (Figure 5, left). While the DDSP Inference module faithfully re-synthesizes the ground-truth audio, MIDI-DDSP generates a coherent performance from a series notes, and has rich, varying expressions across the notes. MIDI2Params failed to generate coherent expressions within a note, generating unrealistic pitch and loudness contours. Also, MIDI2Params stopped the note in the middle when generating the fifth note, suggesting that modelling expressive performance only in synthesis level is limited in long-term coherence even inside a single note. On the contrary, the note expression modeling in MIDI-DDSP allows it to model dependencies at the granularity of the note sequence and use synthesis parameters to model the frame-wise parameter changing inside a single note. The two concatenative synthesizers Ableton and FluidSynth generate the same note expression with identical vibrato and volume for all notes. Although the expression is coherent inside a single note, they fail to generate expression dependencies between different notes automatically.

### 4.4 Effects of Note Expression Controls

To evaluate the behavior of the note expression controls, we measure how well each control correlates with itself after a roundtrip through synthesis. That is, for each sample in the test set, we interpolate the control from lowest (0) to highest (1) in an interval of 0.1 and generate synthesis parameters. Then we extract the note expressions from these synthesis parameters. Table 2 reports the correlation between the value we put in and the value observed after synthesis. All controls exhibit

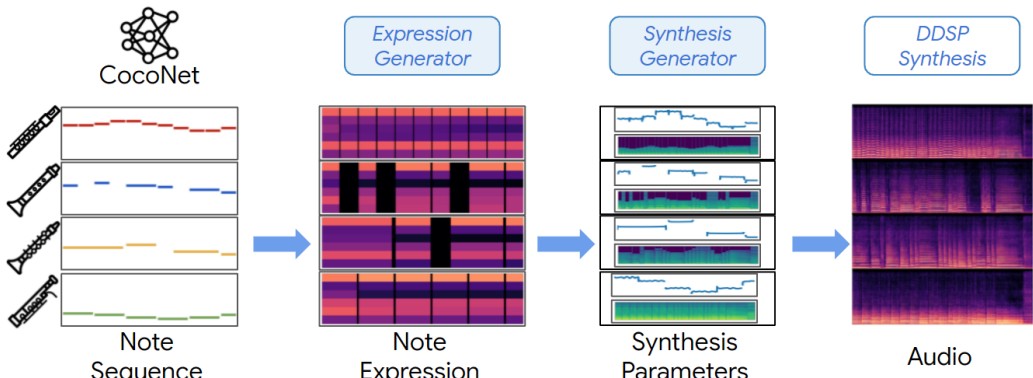

Figure 6: MIDI-DDSP can take input from different sources (human or other models) by designing explicit latent representations at each level. A full hierarchical generative model for music can be constructed by connecting MIDI-DDSP with an automatic composition model. Here, we show MIDI-DDSP taking note input from a score level Bach composition model and automatically synthesizing a Bach quartet by generating explicit latent for each level in the hierarchy.

strong correlation as desired, except for volume peak position. A low correlation may stem from characteristics of the instrument, or imbalances of those performance techniques in the dataset.

Figure 4 illustrates how each note expression affects properties of the sound. For example, as we increase vibrato, we see stronger fluctuations in pitch. Similarly, changing the volume peak position changes the shape of the amplitude curve.

### 4.5 FINE GRAINED CONTROL OR FULL END-TO-END GENERATION

The structured modelling approach of MIDI-DDSP enables end users to have as much or as little control over the output as they want. A user can add manipulations at certain levels of the hierarchy or let the model guide the synthesis.

On one end of this spectrum, Figure 2 shows the results of an end user manipulating each level of MIDI-DDSP. Because different levels of the MIDI-DDSP hierarchy correspond with different musical attributes, a user can make manipulations at the note-level to change the attack noise and volume to create staccato notes (second green box in Figure 2) or a user could make adjustments to the synthesis-level to control the pitch contour for making a "pitch bend" (yellow box in Figure 2).

On the other end of the spectrum, MIDI-DDSP can be paired with generative symbolic music models to make fully generated, realistic end-to-end performances. As shown in Figure 6, MIDI-DDSP can be combined with a composition Bach chorales model COCONET (Huang et al., 2017), to form a fully generated musical quartet that sounds like real instruments performance. Readers are encouraged to listen to both the hand-tuned and end-to-end performances on our accompanying website.

## 5 CONCLUSION

We proposed MIDI-DDSP, a hierarchical music modeling system that factorizes the generation of audio to note, performance, and synthesis levels. By proposing explicit representations for each level alongside modeling note expression, MIDI-DDSP enables effective manipulation and realistic automatic generation of music. We show, experimentally, that the input controls for MIDI-DDSP are correlated with desired performance characteristics (e.g., vibrato, volume, etc). We also show that listeners preferred MIDI-DDSP over existing systems, while enabling fine-grained control of these characteristics. MIDI-DDSP can also connect to other models to construct a full audio generation model, where beginners can obtain realistic novel music from scratch, while expert users can manipulate results based on model prediction to realize unique musical design.

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

# A    APPENDIX

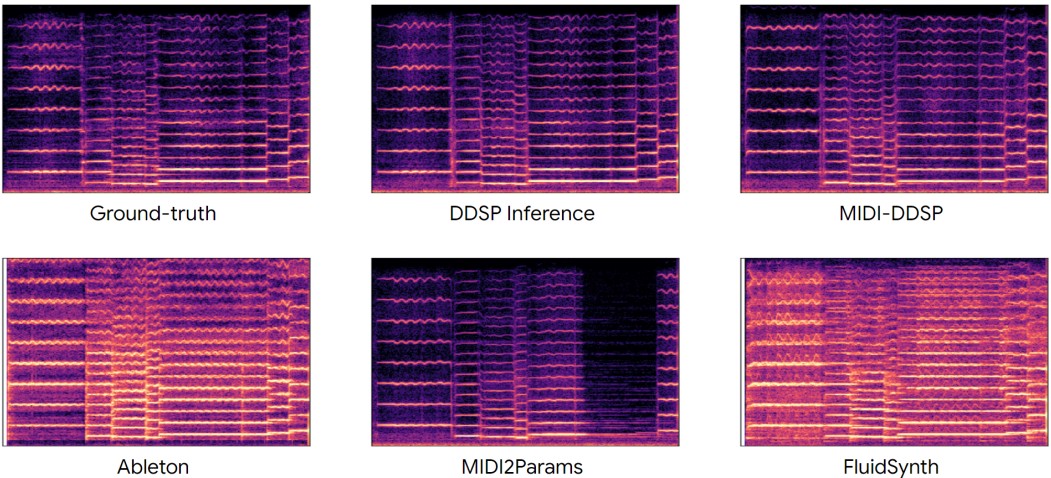

Figure 7: The enlarged log-scale Mel spectrograms of synthesis results in Figure 5

# B    MODEL DETAILS

## B.1    DDSP SYNTHESIZER

Differentiable Digital Signal Processing (DDSP) (Engel et al., 2020a) enables differentiable audio synthesis by using a harmonic plus noise model (Serra & Smith, 1990). The harmonic signal is synthesized using a bank of $K_h$ sinusoidal oscillators parameterized by fundamental frequency $f_0(t)$, harmonic amplitudes $a(t)$, and harmonic distribution $\boldsymbol{h}(t)$. The noise signal is synthesized by a filtered noise synthesizer parameterized by noise magnitudes $\boldsymbol{\eta}(t)$. Due to the strong inductive bias introduced by DDSP, the synthesis parameters are highly interpretable, i.e., opposed to some high-dimensional learned latent vector producing audio, with DDSP models the network's output is input to the harmonic plus noise model, whose parameters are interpretable by definition. For the DDSP synthesis used in this work, $\boldsymbol{s}(t) = (f_0(t), a(t), \boldsymbol{h}(t), \boldsymbol{\eta}(t))$, where $f_0(t) \in \mathbb{R}^{1 \times t}$, $a(t) \in \mathbb{R}^{1 \times t}$, $\boldsymbol{h}(t) \in \mathbb{R}^{60 \times t}$, $\boldsymbol{\eta}(t) \in \mathbb{R}^{65 \times t}$.

Same as the original DDSP synthesizer, the noise signal is generated by filtering uniform noise given noise magnitudes $\boldsymbol{\eta}(t)$ in $K_n$ frequency bins. An exponential sigmoid nonlinearity is applied to the raw neural network output to generate the $a(t)$, $\boldsymbol{h}(t)$ and $\boldsymbol{\eta}(t)$: exp-sigmoid$(x) = 2.0 \cdot$ sigmoid$(x)^{\log 10} + \epsilon$, where $\epsilon = 10^{-7}$. The harmonic distribution $\boldsymbol{h}(t)$ is further normalized to constrain amplitudes of each harmonic component: $\sum_{k=0}^{K} h^k(t) = 1$.

In this paper, we use 60 harmonic bins to synthesize harmonic signal, and 65 magnitude bins to synthesize filtered noise. The frame size is set to 64 samples per frame, and the sample rate of the audio synthesis is set to 16000 Hz. After $x(t)$ is synthesized, a reverb module is applied to $x(t)$ to capture essential reverberation in the environment and instrument. The reverb module is implemented as a frequency domain convolution with a learnable impulse response parameter. This paper uses different reverb parameters for different instruments, and the reverb parameters are learned with back-propagation. The learnable impulse response of the reverb is set to have a length of 48000 sample points. In experiments, we found the latter part of the impulse response, which has minimal impact on the timbre and environmental reverb, would cause a very long lingering sound. Thus, in inference time, we add an exponential decay to the impulse response after 16000 samples to constrain the lingering effect:

$$\left.\begin{array}{ll} \text{IR}'(t) = \text{IR}(t), & 0 \le t \le 16000 \\ \text{IR}'(t) = \text{IR}(t) \cdot \exp\left(-4(t - 16000)\right), & 16000 < t \le 48000 \end{array}\right\} \tag{3}$$

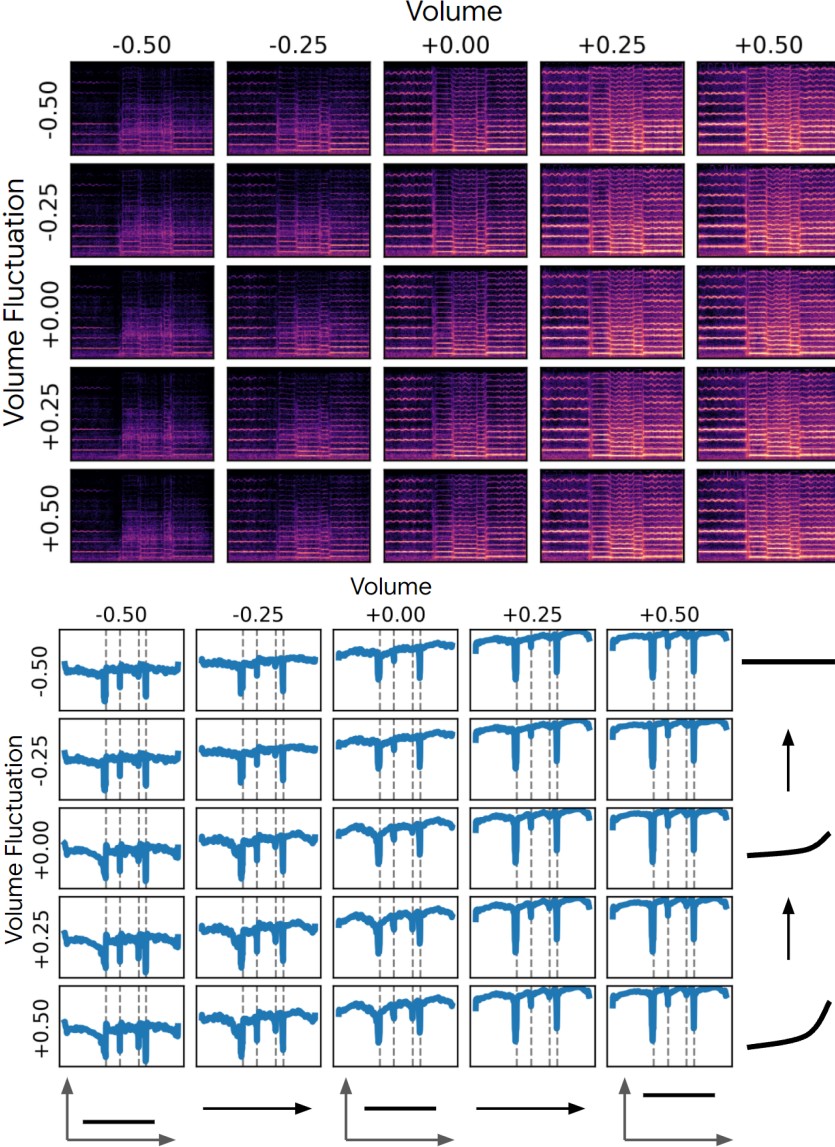

Figure 8: The effect of modifying the 'Volume' and 'Volume Fluctuation' note expression for a sample. Each row shows the amplitudes of the notes when fixing 'Volume Fluctuation' and changing 'Volume', while each column shows the amplitudes of the notes when fixing 'Volume' and changing 'Volume Fluctuation'. The number indicates the amount of modification to the note expression control where (+0, +0) is the original sample. Upper figure shows the spectrogram of generations, and the bottom figure shows the amplitude of the generations. The cartoon on the side indicates how the modified feature would change the synthesis parameters. The gray dash line in the bottom figure indicates the note boundaries.

where $\mathrm{IR}(t)$ is the original impulse response, and $\mathrm{IR}'(t)$ is the impulse response after decay.

## B.2 DDSP INFERENCE

In work proposed by Engel et al. (2020a), an autoencoder is built to reconstruct audio by predicting synthesis parameters from audio features. We refer to this prior work as the "DDSP autoencoder." Given an audio, fundamental frequency is estimated by CREPE (Kim et al., 2018) model, and the loudness is extracted via an A-weighting of the power spectrum (Hantrakul et al., 2019). Then, the

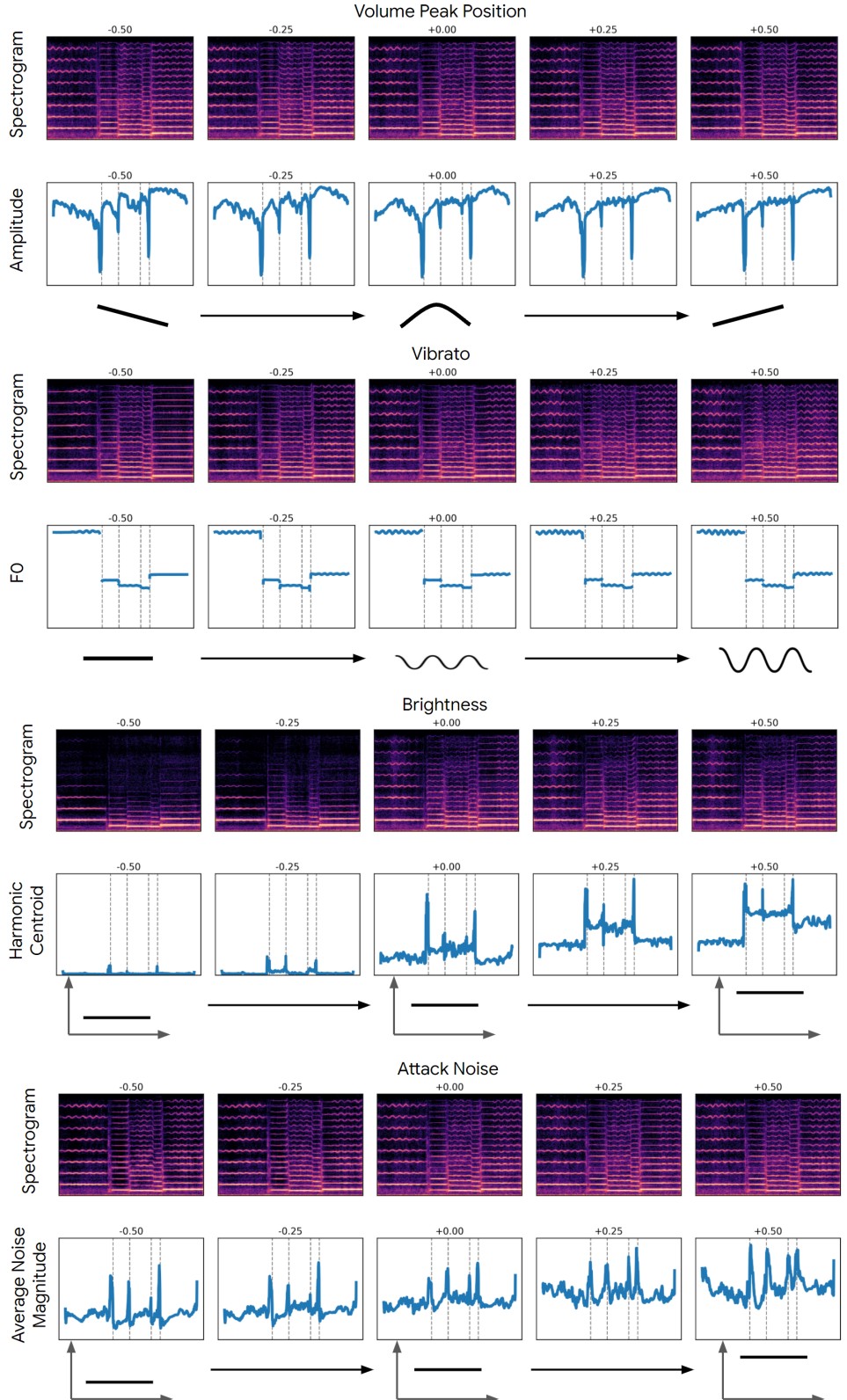

Figure 9: The effect of modifying different note expression parameters on an existing sample. The number indicates the amount of modification to the note expression control where (+0) is the original sample. Upper figure shows the spectrogram of generations, and the bottom figure shows the quantity of the generations affected by the control. The cartoon on the side indicates how the modified feature would change the synthesis parameters. The gray dash line in the bottom figure indicates the note boundaries.

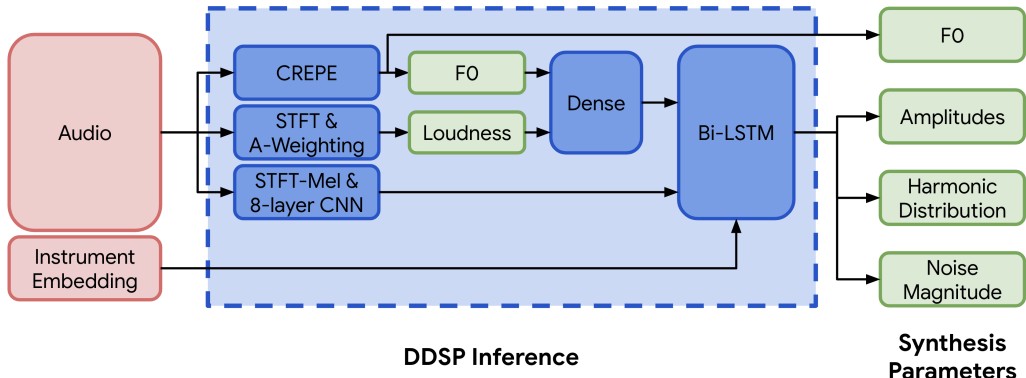

Figure 10: The architecture of the DDSP Inference. The DDSP Inference module extract $f_0$ and loudness from audio, and use an 8-layer CNN to extract features from Mel-spectrogram. A bi-directional LSTM takes in all features from audio and predict synthesis parameters.

fundamental frequency and loudness are input to a multi-layer perceptron (MLP) respectively. The output of the MLPs are concatenated and passed to a uni-directional GRU. Finally, an MLP is used to predict synthesis parameters from GRU output.

Our DDSP Inference module differs from the above DDSP autoencoder by enabling more information extracted from audio input. The architecture of the DDSP Inference is shown in Figure 10. In addition to fundamental frequency estimated by CREPE and loundess extracted via A-weighting of the power spectrum, an 8-layer convolutional neural network (CNN) (Lecun et al., 1998) is used to extract features from log-scale Mel-spectrogram, and a bi-directional long-short term memory network (LSTM) (Hochreiter & Schmidhuber, 1997) is then applied to extract contextual features. The use of CNN applied on log-scale Mel-spectrogram help model to extract more information from audio input, thus enabling more accurate synthesis parameter estimation.

In our DDSP Inference module, a fully-connected network is applied on fundamental frequency and loudness. The output is concatenated with the output of CNN, send to the bi-directional LSTM to extract contextual features. Another fully connected layer are used to map the features to synthesis parameters. The CNN network in the DDSP Inference module is similar to the one used by Kong et al. (2020) for computational efficiency. The detailed architecture of the CNN is shown in Table 3.

The DDSP Inference Module is optimized via Adam optimizer in a batch size of 16 and a learning rate of $3e - 4$. We choose a log-scale Mel-spectrogram with 64 frequency dimensions in this work to extract features by CNN. The features extracted by CNN are mapped to 256 dimensions, and concatenated to the features extracted from the fundamental frequency and loudness. When trained in the multi-instrument setting, a 64 dimensional instrument embedding is also concatenated to the aforementioned feature vectors. The bi-directional LSTM has a hidden dimension of 256.

### B.3 Expression Controls

Given frame-wise extracted synthesis parameters $s(t)$ for the $i$th note, $n_i$, in a note sequence, we say that the $i$th note starts at frame $T_{on}$ and ends at frame $T_{off}$. We define $\tau \in [T_{on}, T_{off}]$ to be every frame that a note is active and the total frame duration of the note is $T_n = T_{off} - T_{on}$. The synthesis parameters for a the whole duration of the note are defined by a fundamental frequency $f_0(\tau)$ contour, an amplitude contour $a(\tau)$, a harmonic distribution $h(\tau)$, and a set of noise magnitudes $\eta(\tau)$.

The amplitude contour, $a(\tau)$, and noise magnitudes $\eta(\tau)$ are transformed into dB scale:

$$a'(\tau) = 20 \log_{10} a(\tau), \quad \eta'(\tau) = 20 \log_{10} \eta(\tau) \tag{4}$$

The note expression controls are extracted as follows:

| ConvBlock | Kernel Size | Stride | Filter Size |
|---|---|---|---|
| Conv2d | (3,3) | (1,1) | $K_{Filters}$ |
| Batch norm + ReLU | - | - | - |
| Conv2d | (3,3) | (1,1) | $K_{Filters}$ |
| Batch norm + ReLU | - | - | - |
| Average pooling | (1,2) | - | - |

| Mel-CNN | Output Size | Filter Size | Dropout Rate |
|---|---|---|---|
| LogMelSpectrogram | (1000, 64, 1) | - | - |
| ConvBlock | (1000, 32, 64) | 64 | - |
| Dropout | (1000, 32, 64) | - | 0.2 |
| ConvBlock | (1000, 16, 128) | 128 | - |
| Dropout | (1000, 16, 128) | - | 0.2 |
| ConvBlock | (1000, 8, 256) | 256 | - |
| Dropout | (1000, 8, 256) | - | 0.2 |
| ConvBlock | (1000, 4, 512) | 512 | - |
| Reshape | (1000, 2048) | - | - |
| Dense | (1000, 256) | 256 | - |

Table 3: The architecture of the Mel-CNN (bottom) used to extract features from log-scale Mel-spectrogram in the DDSP Inference module. The Mel-CNN uses convolutinal blocks defined in the first table below.

**Volume** is the sum of the amplitude contour (in dB) normalized over the length of the note, i.e., the mean amplitude over the note,

$$\frac{1}{T_n} \sum_{i=1}^{T_n} a'(i). \tag{5}$$

**Volume fluctuation** measures the standard deviation of the amplitude curve (in dB):

$$\sqrt{\frac{1}{T_n} \sum_{i=1}^{T_n} (a'(i) - \overline{a'}(\tau))^2}. \tag{6}$$

where $\overline{a'}(\tau)$ is the mean amplitude over the whole note.

**Volume peak position** is the location (in time) of the highest amplitude value normalized over the length of the note,

$$\frac{1}{T_n} \arg\max_i a'(i) \quad \forall i \in [1, T_n]. \tag{7}$$

**Vibrato** is inspired by previous works on expressive performance analysis (Marchini et al., 2014; Li et al., 2015; Yang et al., 2016). The vibrato is calculated by applying Discrete Fourier Transform (DFT) to the fundamental frequency sequence:

$$\max_i \mathcal{F}\{f_0(t)\}_i, \tag{8}$$

where $\mathcal{F}\{\cdot\}$ defines the DFT function. Only notes with a vibrato rate between 3 to 9 Hz and longer than 200ms are recorded. Otherwise, the vibrato of the note is set to 0. In calculating the DFT function, the fundamental frequency, $f_0(t)$, is zero-padded to 1000 frames.

**Brightness** is defined as the spectral centroid (in bin numbers) of the harmonic distribution,

$$\frac{1}{T_n} \sum_{i}^{T_n} \sum_{k=1}^{|\boldsymbol{h}|} k \cdot \boldsymbol{h}^k(i), \tag{9}$$

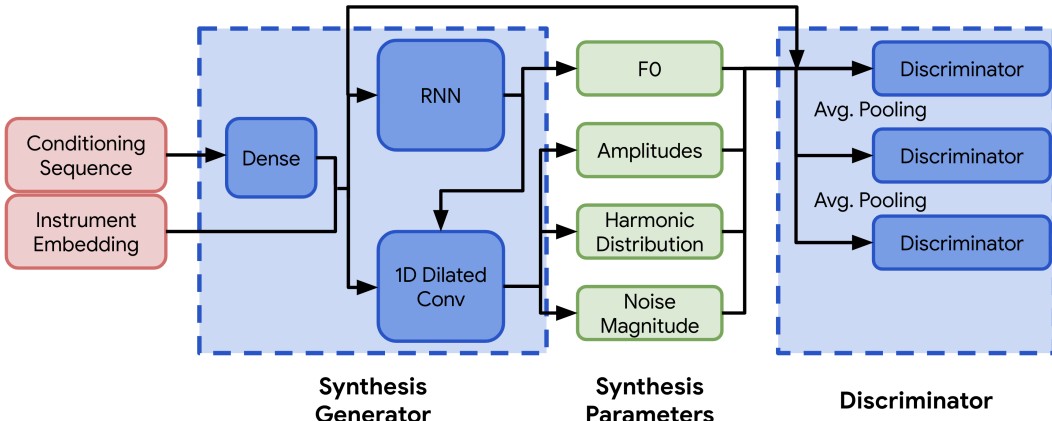

Figure 11: The architecture of the Synthesis Generator. The Synthesis Generator is a GAN whose generator (left) takes in per-note Expression Controls and instrument embedding as a conditioning sequence (red box, left) and produces DDSP synthesis parameters, i.e., $f_0$, Amplitudes, Harmonic Distribution, and Noise Magnitudes (green boxes, middle). These synthesis parameters get turned into audio by the DDSP modules.

where $\boldsymbol{h}^k(i)$ represents the $k$-th bin of the harmonic distribution, $\boldsymbol{h}(\tau)$ in the $i$-th time-step, and we use $|\boldsymbol{h}|$ to refer to the number of bins in the harmonic distribution used by the DDSP module (we use $|\boldsymbol{h}| = 60$, see Appendix B.2).

**Attack Noise** refers to the amount of noise that occurs at the beginning of a note. Many instruments have a high amount of noise in the first few milliseconds of a note (e.g., a bow scraping across a violin string), before the harmonic components are heard. We determine attack noise like

$$\frac{1}{N} \sum_{i=1}^{N} \sum_{k=1}^{|\boldsymbol{\eta}|} \boldsymbol{\eta}'^k(i), \tag{10}$$

where $N$ determines how many of the first few frames we look at to determine the attack noise (we set $N = 10$, corresponding to 40ms), $\boldsymbol{\eta}'^k(i)$ represents the $k$-th bin of the dB-scaled noise magnitudes in the $i$-th time-step, and $|\boldsymbol{\eta}|$ is the number of noise magnitude bins in the DDSP module (set to $|\boldsymbol{\eta}| = 65$, see Appendix B.2).

Recall that all expression controls are normalized to be in the interval $[0.0, 1.0]$, concatenated to a 6-dimensional vector, and repeated for the full frame duration of the note, $T_n$, before we use them with the MIDI-DDSP networks. See Sections 3.2 and 3.3 for additional information. We note that these are not the only ways to determine the "expression" of a note, nor are our definitions definitive. Expression does not even need to be hand designed, and could perhaps be learned in an unsupervised way by neural networks. However, we designed our expression controls because they are all inherently interpretable (compared to black box neural net features). We leave the exploration of other ways to define expression for future work.

For constructing a conditioning sequence from these note expressions and an input note sequence, the frame-wise note expressions and note pitch is expanded to a note-length sequence by repeating these parameters for the number of frames the note occupies. Then, these note expression and pitch parameters are concatenated with note onsets and offsets (a binary flag at the start and end of a note, respectively), and a scalar note positioning code to provide additional information about note boundaries.

### B.4 SYNTHESIS GENERATOR

The Synthesis Generator is a Generative Adversarial Network (GAN) Goodfellow et al. (2014) whose generator takes in the per-note Expression Controls, and produces the DDSP synthesis parameters. The architecture of the Synthesis Generator is shown in Figure 11. The Synthesis Generator

itself contains two networks: an autoregressive RNN, and a stack of dilated 1-D convolutions. The autoregressive RNN that generates an $f_0$ curve based on a note sequence (e.g., MIDI) and the corresponding Note Expression Controls (described in Section B.3). The CNN stack then generates an amplitude curve, a harmonic distribution, and noise magnitudes which are passed to the DDSP modules for synthesis. The generated $f_0$ curve, amplitude curve, harmonic distribution, and noise magnitudes are all passed to a discriminator.

**$f_0$ Generation using the Autoregressive RNN** The autoregressive RNN for $f_0$ generation consists of a single-layer Bi-LSTM and a 2-layer GRU that autoregressively samples the note's exact pitch, by predicting $f_0$ frequency offset (in units of semitones) with respect to a known $f_0$ MIDI note number for the current MIDI note. For example, if at some frame the note specified by the note sequence is A4, which has a MIDI number[2] of $f_0^{\text{A4}} = 69$, and the ground truth $f_0$ in the audio is $f_0^{\text{GT}} = 69.2$, the autoregressive RNN is expected to predict $f_0^{\text{GT}} - f_0^{\text{A4}} = 0.2$, indicating that the $f_0$ at the current frame should be 0.2 semitones above the $f_0$ determined by the current MIDI note, $f_0^{\text{A4}} = 69$. The autoregressive RNN outputs a categorical distribution of pitch offsets in the range of $[-1.00, 1.00]$ semitones, quantized to 201 bins, where each bin represents 0.01 semitone. The final frequency, $f_0$, input to DDSP synthesizer is converted from semitone units to Hz using $f(n) = 440 \cdot 2^{(n-69)/12}$, where $f$ is the frequency in Hz, and $n$ is the integer MIDI note number. Both the single-layer Bi-LSTM and a 2-layer GRU in the Synthesis Generator have a hidden dimension of 256. At inference time, the pitch offset is sampled using nucleus sampling (Holtzman et al., 2020) with $p = 0.95$ to avoid sudden unrealistic change in fundamental frequency contour. Similar approaches for using autoregressive RNNs to sample $f_0$ curves has been proposed for speech synthesis and conversion (Wang et al., 2018b; Morrison et al., 2020).

**Generating the Rest of the Synthesis Parameters** A stack of dilated 1-D convolution layers, *a la* WaveNet (van den Oord et al., 2016), is used to generate the rest of synthesis parameters, i.e., the base amplitude for the note, $a(t)$, the harmonic amplitudes, $\boldsymbol{h}(t)$, and the noise magnitudes $\boldsymbol{\eta}(t)$. This network uses the predicted $f_0(t)$, and concatenated expression control vector, $\boldsymbol{c}(t)$, as conditioning. This network consists of 4 convolutional stacks, with each stack having five layers of 1-D convolution with exponentially increasing dilation rate followed by ReLU activation (Maas et al., 2013) and layer normalization (Ba et al., 2016). For clarity, we refer to this network as the "Dilated CNN."

The conditioning sequence input is mapped to a 256-dimensional vector by a linear layer before being sent into an autoregressive RNN and dilated convolution network. If trained on the multi-instrument dataset, a 64-dimension instrument embedding is concatenated after the linear layer. A dropout of rate 0.5 is applied to all GRU units, and during training the input is teacher-forced to avoid overfitting and exposure bias.

The full details of the dilated convolutional network architecture are shown in Table 4.

**Discriminator for the Synthesis Generator** The architecture of the discriminator used with the Synthesis Generator is shown in Figure 12, and the detailed architecture of each discriminator block is shown in Table 5. The discriminator is motivated by the multi-scale discriminator network in previous works generating waveforms and spectrograms (Kumar et al., 2019; Lee et al., 2021a). It consists of 3 discriminator networks with same architecture that take input signals with different downsample rate in an effort to learn the features of synthesis parameters at different time resolutions. Each discriminator network consists of 4 blocks at each scale, and each block extracts features from predicted synthesis parameters and the conditioning sequence. For each feature stream in a block, two 1-D convolutional layers are used with Leaky-ReLU activation function (Xu et al., 2015), with skip connections and layer normalization (Ba et al., 2016).

**Synthesis Generator Losses** The Synthesis Generator is trained by minimizing both reconstruction loss and adversarial loss:

$$\mathcal{L} = \mathcal{L}_{recon} + \alpha \mathcal{L}_{adv} = (\mathcal{L}_{CE(f_0)} + \mathcal{L}_{spec}) + \alpha \mathcal{L}_{adv}, \tag{11}$$

---

[2]https://www.inspiredacoustics.com/en/MIDI_note_numbers_and_center_frequencies

| Dilated Stack | Kernel Size | Dilation Rate | Stride | Filter Size |
|---|---|---|---|---|
| Conv1d | 3 | 1 | 1 | $K_{Filters}$ |
| ReLU + layer norm | - | - | - | - |
| Add residual | - | - | - | - |
| Conv1d | 3 | 2 | 1 | $K_{Filters}$ |
| ReLU + layer norm | - | - | - | - |
| Add residual | - | - | - | - |
| Conv1d | 3 | 4 | 1 | $K_{Filters}$ |
| ReLU + layer norm | - | - | - | - |
| Add residual | - | - | - | - |
| Conv1d | 3 | 8 | 1 | $K_{Filters}$ |
| ReLU + layer norm | - | - | - | - |
| Add residual | - | - | - | - |
| Conv1d | 3 | 16 | 1 | $K_{Filters}$ |
| ReLU + layer norm | - | - | - | - |
| Add residual | - | - | - | - |

| Dilated CNN | Output Size | Kernel Size | Dilation Rate | Stride | Filter Size |
|---|---|---|---|---|---|
| Conditioning Sequence | (1000, 384) | - | - | - | - |
| Conv1d | (1000, 128) | 3 | 1 | 1 | 128 |
| Dilated Stack | (1000, 128) | 3 | - | 1 | 128 |
| Dilated Stack | (1000, 128) | 3 | - | 1 | 128 |
| Dilated Stack | (1000, 128) | 3 | - | 1 | 128 |
| Dilated Stack | (1000, 128) | 3 | - | 1 | 128 |
| Layer norm | (1000, 128) | - | - | - | - |
| Dense | (1000, 126) | - | - | - | 126 |

Table 4: The architecture of dilated convolution network (bottom) used in the Synthesis Generator to generate amplitude, harmonic distribution and noise magnitude. The dilated convolution network uses dialted stack layers defined in the first table below.

| Discriminator Block - conditioning sequence | Output Size | Kernel Size | Stride | Filter Size |
|---|---|---|---|---|
| Conditioning Sequence | $(T_{in}, D_c)$ | - | - | - |
| Conv1d | $(T_{in}/2, 256)$ | 3 | 2 | 256 |
| Add output from Discriminator Block of synthesizer parameters | $(T_{in}/2, 256)$ | - | - | - |
| Leaky ReLU | $(T_{in}/2, 256)$ | - | - | - |
| Add residual and layer norm | $(T_{in}/2, 256)$ | - | - | - |

| Discriminator Block - synthesis parameters | Output Size | Kernel Size | Stride | Filter Size |
|---|---|---|---|---|
| synthesis parameters | $(T_{in}, D_s)$ | - | - | - |
| Conv1d | $(T_{in}/2, 256)$ | 3 | 2 | 256 |
| Leaky ReLU | $(T_{in}/2, 256)$ | - | - | - |
| Add residual and layer norm | $(T_{in}/2, 256)$ | - | - | - |

Table 5: Details of the discriminator blocks used in the Synthesis Generator.

where $\mathcal{L}_{recon}$ is the reconstruction loss, $\mathcal{L}_{adv}$ is the adversarial loss, and $\alpha$ is a hyperparameter that controls the weight of the adversarial loss term. The reconstruction loss, $\mathcal{L}_{recon}$, consists of two pieces: a cross-entropy loss, $\mathcal{L}_{CE(f_0)}$, on the $f_0$ to train the autoregressive RNN, and a multi-scale spectral loss, $\mathcal{L}_{spec}$ to train the Dilated CNN.

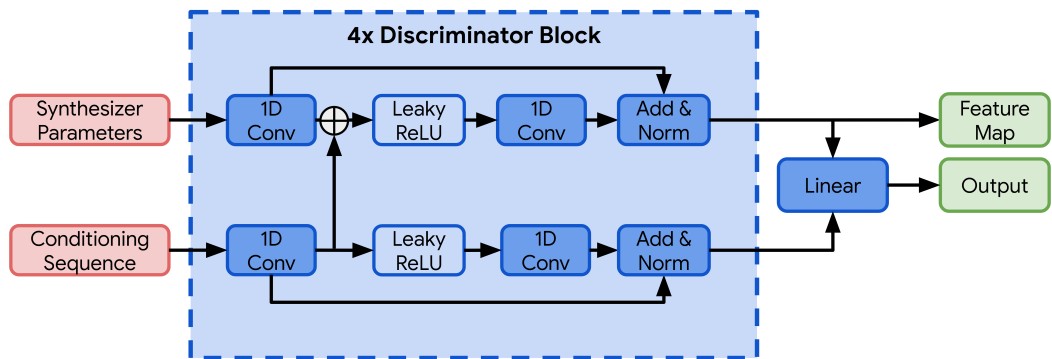

Figure 12: The architecture of the discriminator used in the Synthesis Generator.

The cross-entropy loss for the autoregressive RNN is defined as

$$\mathcal{L}_{CE(f_0)} = -\sum_i f_{0_i} \log \hat{f}_{0_i}, \tag{12}$$

where $i$ is the length of the entire sequence in frames, $f_0$ is the ground truth fundamental frequency curve, and $\hat{f}_0$ is the estimated fundamental frequency curve.

The multi-scale spectral loss, $\mathcal{L}_{spec}$, is used for the reconstruction loss. This loss is used for reconstruction in the original DDSP paper (Engel et al., 2020a). The multi-scale spectral loss computes the L1 difference between the magnitude spectrogram of the predicted and target audio by comparing the computed spectrograms at a number of different FFT sizes. Given the magnitude spectrogram of the predicted audio $\hat{S}_i$ and that of the target audio $S_i$ with FFT size $i$, the multi-scale spectral loss computes the $L1$ difference between $\hat{S}_i$ and $S_i$ as well as $\log \hat{S}_i$ and $\log S_i$:

$$\mathcal{L}_{spec}^{(i)} = ||S_i - \hat{S}_i||_1 + \beta || \log S_i - \log \hat{S}_i||_1, \tag{13}$$

$$\mathcal{L}_{spec} = \sum_i \mathcal{L}_{spec}^{(i)} \quad \forall i \in \{2048, 1024, 512, 256, 128, 64\}. \tag{14}$$

The adversarial loss, $\mathcal{L}_{adv}$, used to train the dilated CNN in the Synthesis Generator. This loss combines a least-squares GAN (LSGAN) (Mao et al., 2017) objective and a feature matching objective objective (Kumar et al., 2019):

$$\mathcal{L}_{adv} = \mathcal{L}_{lsgan} + \gamma \mathcal{L}_{fm}. \tag{15}$$

Given discriminator network $D_k$ in $k$-th scale, the LSGAN objective to train the Synthesis Generator can be written as

$$\mathcal{L}_{lsgan} = \mathbb{E}_c \left[ \sum_k ||D_k(\hat{s}, c) - 1||_2 \right], \tag{16}$$

and the LSGAN objective for training the discriminator is given by

$$\min_{D_k} \mathbb{E} \left[ ||D_k(s, c) - 1||_2 + ||D_k(\hat{s}, c)||_2 \right], \forall k. \tag{17}$$

In this work, we use 3 scales, so $k = [1, 2, 3]$. Given the output of $i$-th feature map from one of the 4 layers of the $k$-th discriminator $D_k$, the feature map matching objective is calculated as L1 difference between corresponding feature map:

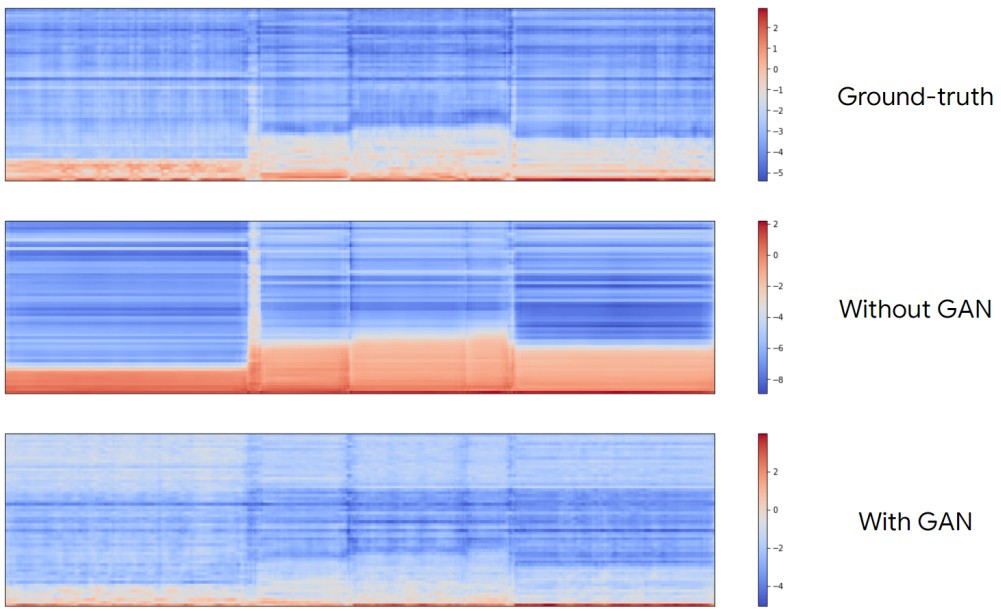

Figure 13: The effect of GAN in overcoming the "over-smoothing" problem in harmonic distribution generation. From top to bottom: (top) the ground-truth harmonic distribution of the test-set sample, (middle) the harmonic distribution of the same sample predicted without a discriminator used in training, (bottom) the harmonic distribution predicted with discriminator and adversarial training.

$$\mathcal{L}_{fm} = \mathbb{E}_{\boldsymbol{s},\boldsymbol{c}} \left[ \sum_{i=1}^{4} \frac{1}{N_i} ||D_k^{(i)}(\boldsymbol{s}, \boldsymbol{c}) - D_k^{(i)}(\hat{\boldsymbol{s}}, \boldsymbol{c})||_1 \right], \tag{18}$$

where $N_i$ is the number of units in $i$-th layer of the feature map.

In training, we use a stop gradient between the adversarial loss and the autoregressive RNN. That is, the RNN is trained by the cross-entropy loss only. The Synthesis Generator is trained using 4 seconds of audio with a frame length of 4ms, resulting in a sequence length of 1000. The Synthesis Generator is optimized via Adam optimizer in a batch size of 16 and a learning rate of 0.0003, with an exponential learning rate decay at a rate of 0.99 per 1000 steps. The discriminator is optimized using Adam optimizer in a batch size of 16 and a learning rate of 0.0001. $\alpha = 1$, $\beta = 1$, and $\gamma = 10$ are used for loss coefficients.

### B.5    ON THE IMPROVEMENT OF USING GAN FOR THE SYNTHESIS GENERATOR

Without GAN training, the Synthesis Generator will suffer from the "over-smoothing" problem where the model generates uniform and smoothed harmonic distribution and noise magnitudes over the whole note. Figure 13 shows this effect qualitatively. With the introduction of adversarial training, the proposed model overcomes the over-smoothing problem potentially caused by one-to-many mapping.

### B.6    EXPRESSION GENERATOR

The Expression Generator creates a set of Expression Controls (see Section B.3) given an input note sequence. The architecture of the Expression Generator is shown in Figure 14. The Expression Generator consists of two parts: a single-layer bidirectional GRU extracts context information from input, and a two-layer autoregressive GRU generates note expression.

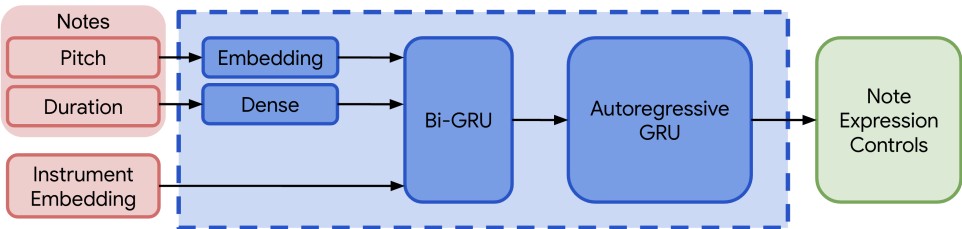

**Expression Generator**

Figure 14: The architecture of the Expression Generator. The Expression Generator consists of two parts: a single-layer bidirectional GRU extracts context information from input, and a two-layer autoregressive GRU generates note expression. The input to the bi-directional GRU is a concatenation of pitch embedding vector, duration feature vector and instrument embedding.

In the input to the Expression Generator, the discrete pitch is mapped into a feature vector of 64 dimensions via an embedding layer, and the scalar duration is mapped into a feature vector of 64 dimensions via a fully-connected layer. The instrument embedding input is a 64 dimension feature vector. The input to the bi-directional GRU is a concatenation of pitch embedding vector, duration feature vector and instrument embedding. The bi-directional GRU and auto-regressive GRU all use a hidden size of 128 and a dropout rate of 0.5. Input dropout with a rate of 0.5 is applied to the teacher-forced input in training time to avoid over-fitting and exposure bias. The output layer of the Expression Generator is a two-layer Multi-layer Perceptron (MLP), which consists of a fully connected layer with a layer normalization before the ReLU nonlinearity used in Engel et al. (2020a).

Data augmentation is applied in the training of the Expression Generator. For the input note sequence, we randomly transpose $\{-3, -2, -1, 0, 1, 2, 3\}$ semitone(s) and randomly stretch the duration by a factor of $\{0.9, 0.95, 1, 1.05, 1.1\}$.

The Expression Generator is trained on a sequence length of 64 notes and a batch size of 256. Adam optimizer (Kingma & Ba, 2014) is used in training with a learning rate of 0.0001. For training the Expression Generator, we use the mean-square loss (MSE) between the estimated Expression Controls and the ground truth Expression Controls (see Section B.3).

### B.7 OTHER TRAINING DETAILS

The Expression Generator, Synthesis Generator, and DDSP Inference Module are trained separately. The Expression Generator is trained for 5000 steps, the Synthesis Generator is trained for 40000 steps, and the DDSP Inference Module is trained for 10000 steps.

## C DATASET

The 13 instruments in the URMP dataset are violin, viola, cello, double bass, flute, oboe, clarinet, saxophone, bassoon, trumpet, horn, trombone, tuba. There were originally 14 instruments in URMP dataset, where soprano saxophone and tenor saxophone are regarded as different instrument. In this paper, we regard both the soprano saxophone and tenor saxophone in URMP dataset as saxophone. The recordings in the dataset have a sample rate of 48kHz but we downsampled them to 16kHz to match the sample rate of the DDSP synthesis module. To train the Synthesis Generator and DDSP Inference, we segmented the recordings into 4 seconds clips with 50% overlap.

In the URMP dataset, the solo recordings are part of ensemble pieces. Splitting the same piece played by different instruments into training and test sets can cause data leakage. Thus, we split the dataset based on a random shuffle of the recordings, and then moved pieces post-hoc such that the same piece does not appear in both training and test set. We use solo recordings in piece number [3, 9, 11, 21, 24, 25, 33, 38, 39, 40, 41, 43] in URMP as test set, and the rest of distinct solo recordings in URMP as training set, as there are repeat use of solo recordings among different pieces.

# D    EXPERIMENT DETAILS

## D.1    DETAILS OF BASELINE METHODS

We use the orchestral strings pack in Ableton[3] without any manual adjustment to synthesize the audio from Ableton. For FluidSynth, we used the `FluidR3_GM.sf2`[4] sound font. We reimplemented the MIDI2Param system proposed by (Castellon et al., 2020) by following the official implementation on GitHub[5]. The MIDI2Params system is only trained in the single instrument setting for the listening test experiments as is done in the original paper. However, as a comparison in our multi-instrument scenario, we trained a multi-instrument MIDI2Params model that had an additional instrument embedding concatenated to the model input.

## D.2    DETAILS OF THE LISTENING TEST

In total, we gathered 960 ratings based on 360 pairwise comparisons from 14 participants for our listening test. Participants were presented with two unlabeled audio clips and asked "Which one of the following recordings sound most like a person playing it on a real violin?" Participants were asked to wear headphones, and were able to listen to the audio clips as many times as they pleased before submitting their answers. The participants chose their preference using a 5-point Likert-like scale, with the first option being "Strong preference for Audio Clip 1", the middle option indicating "No preference", and the final option being "Strong preference for Audio Clip 2."

The pairs of unlabeled clips were drawn from the set of [Ground-truth, DDSP Inference, MIDI-DDSP, Ableton, MIDI2Params, Fluidsynth]. Participants were recruited through a Google internal data labeling platform, and were selected based on a pre-screening pilot study to ensure that the participant was able to provide reasonable evaluations of audio recordings. In this pilot study, we filtered out individuals who rated FluidSynth examples as sounding more like a real violin recording than the Ground-truth violin recording. Participants were not screened based on their musical background.

A Kruskal-Wallis H test of the ratings showed that there is at least one statistically significant difference between the models: $\chi^2(2) = 395.35$, $p < 0.01$. The number of wins for each pair comparison and a Wilcoxon signed-rank test for each pair is shown in Table 6.

---

[3]https://www.ableton.com/en/packs/orchestral-strings/
[4]https://member.keymusician.com/Member/FluidR3_GM/index.html
[5]https://github.com/rodrigo-castellon/midi2params

| Pairs | | wins | ties | losses | $p$ value |
|---|---|---|---|---|---|
| Ableton | MIDI2Params | 39 | 1 | 24 | 7.61e-5 |
| Ableton | DDSP Inference | 11 | 0 | 53 | 2.98e-27 |
| Ableton | FluidSynth | 42 | 0 | 22 | 0.0002 |
| Ableton | Ground-truth | 14 | 1 | 49 | 6.62e-26 |
| Ableton | MIDI-DDSP | 5 | 0 | 59 | 5.34e-16 |
| MIDI2Params | DDSP Inference | 8 | 0 | 56 | 5.73e-42 |
| MIDI2Params | FluidSynth | 31 | 0 | 33 | 0.027* |
| MIDI2Params | Ground-truth | 8 | 0 | 56 | 2.43e-40 |
| MIDI2Params | MIDI-DDSP | 8 | 0 | 56 | 7.16e-28 |
| DDSP Inference | FluidSynth | 55 | 0 | 9 | 2.97e-39 |
| DDSP Inference | Ground-truth | 35 | 0 | 29 | 0.024* |
| DDSP Inference | MIDI-DDSP | 44 | 0 | 20 | 6.02e-05 |
| FluidSynth | Ground-truth | 4 | 0 | 60 | 1.00e-37 |
| FluidSynth | MIDI-DDSP | 9 | 0 | 55 | 7.25e-26 |
| Ground-truth | MIDI-DDSP | 44 | 0 | 20 | 0.00018 |

Table 6: A post-hoc comparison of each pair on their pairwise comparisons with each other, using the Wilcoxon signed-rank test for matched samples ("win" means the first item in the pair is selected). $p$ value less than $0.01/15 = 6.67 \times 10^{-4}$ yields a statistically significant difference. Only two pairs are not significantly different (DDSP Inference vs. Ground-truth, MIDI2Params vs. FluidSynth), and are marked with an asterisk (*).

# E    EXPRESSION ATTRIBUTE CONTROLS

| | Volume | Vol. Fluc. | Vol. Peak Pos. | Vibrato | Brightness | Attack Noise |
|---|---|---|---|---|---|---|
| violin | **.99** | **.84** | **.80** | **.86** | **.96** | **.97** |
| viola | **.98** | **.74** | **.70** | **.82** | **.98** | **.97** |
| cello | **.97** | .64 | .54 | **.74** | **.98** | **.94** |
| double bass | **.98** | **.85** | .34 | **.84** | **.99** | **.95** |
| flute | **.99** | **.87** | .48 | .63 | **.90** | **.97** |
| oboe | **.97** | **.79** | **.72** | **.91** | **.87** | **.97** |
| clarinet | **.98** | **.88** | .63 | **.72** | **.88** | **.97** |
| saxophone | **.97** | **.71** | .43 | **.80** | **.94** | **.96** |
| bassoon | **.99** | **.90** | .56 | **.91** | **.99** | **.96** |
| trumpet | **.97** | **.88** | .55 | **.73** | **.92** | **.92** |
| horn | **.97** | **.88** | .41 | **.64** | **.94** | **.95** |
| trombone | **.97** | **.93** | .59 | .52 | **.99** | **.96** |
| tuba | **.98** | **.91** | .15 | .22 | **.98** | **.93** |

Table 7: The Pearson correlation result of all instruments described in Table 2. The Pearson correlation $r$-values are shown in the table, while $p$-values are omitted as in all entries $p < 0.0001$. We consider Pearson $r$-values larger than 0.7 (marked on **bold**) to mean that our input controls are strongly correlated with the output. For Volume, Brightness, and Attack Noise, all instruments show a strong correlation; for Volume Fluctuation, and Vibrato, the majority of instruments sho a strong correlation.

# F    ACKNOWLEDGEMENTS AND OPEN-SOURCE IMAGE ATTRIBUTIONS

We would like to thank Yi Ren for the discussion on Generative Adversarial Nets for audio generation. We would like to thank Shujun Han for providing advice on making and arranging figures in the paper. We would also like to thank Mark Cartwright for his feedback on the paper and fruitful discussions about it.

The icons used throughout the paper are used under the Creative Commons license via the Noun Project. We gratefully acknowledge the following creators of these images:

- Audio by cataicon from the Noun Project.
- bassoon by Symbolon from the Noun Project.
- Clarinet by Symbolon from the Noun Project.
- composer by Pham Duy Phuong Hung from the Noun Project.
- Flute by Symbolon from the Noun Project.
- Neural Network by Ian Rahmadi Kurniawan from the Noun Project.
- oboe by Symbolon from the Noun Project.
- Synthesizer by Jino from the Noun Project.
- Violin by Olena Panasovska from the Noun Project.
- Violinist by Luis Prado from the Noun Project.

