# OpenReview forum: "MIDI-DDSP: Detailed Control of Musical Performance via Hierarchical Modeling"
_ICLR.cc/2022/Conference — ICLR 2022 Oral_

### Official Review · Reviewer_FkaH · 2021-11-02

**Correctness:** 4
**Technical Novelty And Significance:** 3
**Empirical Novelty And Significance:** 4
**Recommendation:** 8
**Confidence:** 4

**Main Review:**

This paper introduces a layer of controllable expression parameters for the audio rendering of MIDI files. The contribution is threefold:
1. The authors introduce heuristics to extract note-wise expression parameters from low-level synthesis parameters. The choice of the controls are relevant to the dataset the authors use (acoustic instruments).
2. They introduce the synthesis generator, allowing to predict the frame-level synthesis parameters, hence bypassing the DDSP decoder that used to predict them from F0 and loudness contours. The introduction of this model allows a better reconstruction error than the original DDSP. Also, it transfers the manipulation of the sound from the mentioned contours (which are frame-level) to note-level expression controls, more expressive and controllable.
3. The smart design of the expression controls allows them to be controlled by humans, hence allowing further manipulation of the synthesis. Each control is well defined, normalized between 0 and 1, and coarse enough (note-level vs. frame-level) so that a user can easily set them to their preferred value, enhancing the customization possibilities. Human manipulation of the controls seem to have the desired effect, as shown by the correlation study in Table 2 and further supported by the audio material.
A few comments:
- In paragraph 3.3, the authors write that the note expression controls are pooled over the duration of the corresponding note. To my understanding, the controls would rather be unpooled, repeated or upsampled, as there are more frames than notes (very clear in B.4).
- The paragraph about the dataset doesn't mention that it also contains the MIDI ground truth. This could be made explicit.
- The paragraph entitled "Expression Generator" in 4.2 features what looks like a residue of an unwanted sentence
- In Appendix B.3, the definition for brightness seems to be lacking a sum over k, and I'm having a hard time trying to understand the multiplication by i rather than k.
- Some of these expression controls naturally lie in [0;1] (e.g. Volume Peak Position). However, it is not clear how others are normalized (e.g. Vibrato). If normalizing constants are precomputed using feature extraction on the whole dataset, it should be written.


**Summary Of The Paper:**

This paper presents a controllable rendering engine for MIDI files, based on the DDSP framework. Given F0 and loudness contour, DDSP can estimate the parameters of a harmonic + noise synthesis model, to render a corresponding audio file. Similar to MIDI2Params, which predicts framewise FO and loudness contours from a MIDI file, MIDI DDSP introduces an intermediate hierarchical level, allowing to control some newly introduced "expression controls". A mapping from MIDI files to "expression controls" is learnt, so that MIDI files can be automatically rendered. However, because these note wise controls are explicit, they also allow human manipulation of the performance rendering.
Influence of the expression controls is assessed with a correlation study, showing that the human manipulation has the expected effect on the generated performance. This quantitative evaluation is further confirmed by convincing audio examples.

**Summary Of The Review:**

Overall, this works is very well written. Everything should appear pretty clear to anyone familiar with the DDSP framework. The principle is simple yet extremely effective, as it improves reconstruction quality and parameters estimation compared to the state of the art. This new control layer represents a lot of added value compared to MIDI2Params, which was already a great addition to the original DDSP framework. In addition to the paper, the website provides very convincing audio examples which further assess the quality of this work.

---

> ### Author Response · Authors · 2021-11-20
> **Author Response to Reviewer FkaH**
>
> We thank you for the review and the generous comments. We've done our best to address your questions with paper revisions and the comments below.
>
> >" In paragraph 3.3, the authors write that the note expression controls are pooled over the duration of the corresponding note. To my understanding, the controls would rather be unpooled, repeated or upsampled, as there are more frames than notes (very clear in B.4)."
>
> Huge thanks for pointing it out! This is indeed a typo, and we have changed the sentence to "Note expression controls are unpooled (repeated) over the duration of the corresponding note to make a conditioning sequence…" in the revised version of the manuscript.
>
> >" The paragraph about the dataset doesn't mention that it also contains the MIDI ground truth. This could be made explicit."
>
> This is indeed an oversight. We have changed the text to now highlight the inclusion of MIDI ground truth to the introduction of the dataset. We changed the introduction to the dataset to "The URMP dataset contains solo performance recordings and their ground truth note boundary of 13 string instruments and wind instruments, which allows us to test generalization to different instruments."
>
> >" The paragraph entitled "Expression Generator" in 4.2 features what looks like a residue of an unwanted sentence"
>
> Thanks! We've fixed them in the revised version of the manuscript.
>
> >" In Appendix B.3, the definition for brightness seems to be lacking a sum over k, and I'm having a hard time trying to understand the multiplication by i rather than k."
>
> Thanks a lot for pointing out the error in the definition! The definition of the brightness should be $\frac{1}{T_n} \sum_i^{T_n} \sum_{k=1}^{60} k\cdot h^k(i)$. We’ve fixed them in the revised version of the manuscript.
>
> >" Some of these expression controls naturally lie in [0;1] (e.g. Volume Peak Position). However, it is not clear how others are normalized (e.g. Vibrato). If normalizing constants are precomputed using feature extraction on the whole dataset, it should be written."
>
> We've added more details to clarify the text in this regard. All the expression controls are linearly scaled by adding an offset and divided by a coefficient. We determine the parameter for the scaling by computing the histogram of expression controls and adjusting the range to fit almost all points in [0,1]. We avoid simply normalizing the expression controls by min-max normalization to [0,1] because the long-tail effect and outlier values will narrow the effective control range, and the value adjustment is less distinguishable for users.
> There is nothing special per-se about this approach, and other scaling methods such as normalizing by subtracting the mean and dividing by standard deviation should also work. We are planning to find a method for automatic note expression control scaling to allow the model to train on arbitrary datasets.

---

### Official Review · Reviewer_Z9iV · 2021-11-02

**Correctness:** 3
**Technical Novelty And Significance:** 3
**Empirical Novelty And Significance:** 3
**Recommendation:** 8
**Confidence:** 3

**Main Review:**

The paper is well-written, so I only have few minor questions.

While reading the paper, I was wondering the three sub-modules are trained simultaneously. If not, can other dataset be used for training each sub-module to enhance robustness or the performance?

For synthesis generator module, I couldn't find some ablation test about the loss function.

The authors argue that the method can be easily extended when multi-instrument transcription model is ready. However, I see there exist many challenges (e.g. the model relies on CREPE in DDSP synthesis part, also the type of an instrument set between transcription part and MIDI-DDSP should be matched.), the tone can be lowered.

In page 3, "Realistic Note Synthesis", "a" is missing in a word 'concatenative'.


**Summary Of The Paper:**

This paper proposes a music performance modeling network using three sub-modules which are expression generator, synthesis generator, and DDSP inference. The idea of using these three sub-modules to create three-level performance modeling (the three-level control values are notes, performance features, and synthesis parameters) is interesting and the result shows that the proposed model can give more ability of control while not harming the overall audio generation performance. It seems the proposed method that having three trainable networks with two hand-crafted methods (actually note detection is not the part of the contribution of this paper, so I didn't count it) gives full pipeline of hierarchical modeling and this is a good research contribution.

**Summary Of The Review:**

The paper opens a more user controllability in music performance modeling and this is a nice contribution. Also, the idea of having rule-based method with trainable networks in a series of a chain seems useful in many other applications.

---

> ### Author Response · Authors · 2021-11-20
> **Author Response to Reviewer Z9iV**
>
> We thank you for the review and the generous comments. We've done our best to address your questions with paper revisions and the comments below.
>
> >" While reading the paper, I was wondering the three sub-modules are trained simultaneously. If not, can other dataset be used for training each sub-module to enhance robustness or the performance?"
>
> Thank you for the thoughtful suggestion! Yes, the three sub-modules in MIDI-DDSP are separately trained (further detailed in Figure 3). There's no reason why one could not use other datasets to train/pre-train each module separately to enhance the robustness or even learn a different style of playing. Hopefully, we can explore this idea in future work.
>
> >" For synthesis generator module, I couldn't find some ablation test about the loss function."
>
> Thanks very much for considering this problem. We have additional qualitative results of this ablation of loss function available here: https://midi-ddsp.github.io/#loss-ablation.  We compare the original MIDI-DDSP trained on multi-scale spectral loss and GAN objective with two variants: "MIDI-DDSP Params Loss" (Loss on synthesis parameters instead of the multi-scale spectrogram, also with GAN loss) and "MIDI-DDSP without GAN" (multi-scale spectrogram loss, but no GAN). We found that "MIDI-DDSP params loss" has a similar sound quality to the original MIDI-DDSP, while MIDI-DDSP without GAN produced worse output quality (duller, less realistic sounds).
>
> Full audio samples can be found here: https://github.com/MIDI-DDSP/MIDI-DDSP.github.io/blob/master/eval_set_sample_for_loss_ablation.zip.
>
> >" The authors argue that the method can be easily extended when the multi-instrument transcription model is ready. However, I see there exist many challenges (e.g. the model relies on CREPE in the DDSP synthesis part, also the type of an instrument set between transcription part and MIDI-DDSP should be matched.), the tone can be lowered."
>
> Many thanks for pointing this out. We agree that the original language overstated the ease of adapting to polyphony. We have softened the language to "It is important to note that the system relies on pitch detection and note detection, so is currently limited to training on recordings of single monophonic instruments. This approach has the potential to be extended to polyphonic recordings via multi-instrument transcription and multi-pitch tracking, which is an exciting avenue to explore for future work."."
>
> >" In page 3, "Realistic Note Synthesis", "a" is missing in a word 'concatenative'."
>
> Thanks! We've fixed them in the revised version of the manuscript.

---

### Official Review · Reviewer_6Ata · 2021-11-03

**Correctness:** 4
**Technical Novelty And Significance:** 3
**Empirical Novelty And Significance:** 4
**Recommendation:** 8
**Confidence:** 5

**Main Review:**

This paper develops a parametric model of musical audio that leverages insight into the music domain to construct meaningful featurizations of music, together with models trained to invert these featurizations, to construct a hierarchical autoencoder with meaningful semantic features that can be used to control the music generation process.

The technical contributions of this paper consist of a semantic "expression" featurization of DDSP parameters (Section 3.2) and two inverse models: (1) a synthesis generator for converting the expression features back to DDSP synthesis parameters and (2) an expression generator for converting MIDI data back to expression features. The features and models are well-described at a high level in the main paper, and the appendix provides a thorough & precise description of the remaining details. The empirical study seems convincing, with comparisons to both academic prior work (MIDIParams) and commercial software (Ableton, FluidSynth). And the demos sound qualitatively good to my ear.

The paper thoroughly relates this work to related work in the field, making many thoughtful connections to both the symbolic & audio generative modeling literature. These connections both help situate this work in the broader literature, and also provide some welcome scaffolding for thinking more broadly about the field of generative music modeling. I also appreciate the connections to the signal processing literature.


One possible criticism of this paper is that, because its focus is on applying machine learning rather than in the development of machine learning techniques, perhaps it would be better suited for a music domain conference (e.g., ISMIR). I strongly support its publication at ICLR: the paper does a thorough job situating itself in the machine learning literature, making it accessible and appealing to the broader machine learning audience.


One question I have is about the choice of 3 stage modeling (notes -> expression -> synthesis -> audio) as opposed to the 2 stage process used by MIDI2Params (notes -> synthesis -> audio). I appreciate that injecting semantically meaningful performance features provides an additional level of control over the generation process. Does this additional stage also account for the performance improvement of MIDI-DDSP over MIDI2Params? Or is this improvement more attributable to better/bigger models?

Also, related to the previous point, I am a little confused by the MIDI2Params comparisons in Table 1. I am under the impression that the expression step is an innovation of this work (Section 3.2). How are you able to make these breakdown comparisons with MIDI2Params for these substages notes -> expression and expression -> synthesis. Have I misunderstood MIDI2Params?

**Summary Of The Paper:**

This paper extends the DDSP generative modeling approach (Engel et al., 2020) to higher levels of abstraction for high-level MIDI -> audio synthesis with intermediate levels of semantic control over the generation process.

**Summary Of The Review:**

This is a well-written, creative paper. The models are interesting. The figures are excellent. The connections to related work are extensive and thoughtful.

---

> ### Author Response · Authors · 2021-11-20
> **Author Response to Reviewer 6Ata**
>
> We thank you kindly for reviewing the manuscript and for your generous comments. We've done our best to address your questions with paper revisions and the comments below.
>
> >”One question I have is about the choice of 3 stage modeling (notes -> expression -> synthesis -> audio) as opposed to the 2 stage process used by MIDI2Params (notes -> synthesis -> audio). I appreciate that injecting semantically meaningful performance features provides an additional level of control over the generation process. Does this additional stage also account for the performance improvement of MIDI-DDSP over MIDI2Params? Or is this improvement more attributable to better/bigger models?"
>
> Thanks very much for considering this problem. To answer this question, we added an additional ablation experiment without using note expressions, where the synthesis generator is trained to directly use the note sequence instead of conditioning on a combination of note sequence and note expressions. This ablation is trained with the same loss and GAN objective and uses the same hyperparameters as MIDI-DDSP. Qualitatively, the ablation audio sounds similarly good as MIDI-DDSP. However, occasionally it will generate notes lacking almost any expression (constant f0 and loudness throughout the whole note) that sound less realistic. (please see https://midi-ddsp.github.io/#expression-ablation for further details).
>
> Thus, we conclude that the expression controls slightly increase the quality of automated synthesis. More importantly, they also enable a user to make edits at the note expression level, and these new controls ultimately enable more natural-sounding synthesis; this type of editing is not an option in MIDI2Params or the ablation.
>
> You can find all experiment result samples at https://github.com/MIDI-DDSP/MIDI-DDSP.github.io/blob/master/no_expression_experiment.zip.
>
> >" Also, related to the previous point, I am a little confused by the MIDI2Params comparisons in Table 1. I am under the impression that the expression step is an innovation of this work (Section 3.2). How are you able to make these breakdown comparisons with MIDI2Params for these substages notes -> expression and expression -> synthesis. Have I misunderstood MIDI2Params?"
>
>
> Given a set of notes, MIDI2Params will generate f0 and loudness contours. Then the DDSP decoder will predict synthesis parameters from those generated f0/loudness curves.
>
> For notes -> expression, we evaluate teacher-forced prediction accuracy of the note expression module in MIDI-DDSP. For MIDI2Params, no such expression controls exist, so for evaluation, we use teacher-forced prediction accuracy in a manner that is as similar to our system as is possible. We feed in frame-wise notes and previous ground-truth frame-wise output (f0 and loudness) to auto-regressively generate the next frames for the next note. After the output of the next note is generated, we feed them into the DDSP decoder as usual to get the synthesis parameter. We then evaluate the note expression control accuracy by extracting them from synthesis parameters generated from MIDI2Params.
>
> For expression -> synthesis, we are evaluating the reconstruction performance of the synthesis generator. MIDI-DDSP has additional conditioning (note expression + notes) compared to MIDI2Params (notes), which makes it not a completely apples-to-apples comparison, but it does fairly highlight the advantage of the three-stage generation over the two-stage model.

---

### Decision · Program_Chairs · 2022-01-20

**Decision:**

Accept (Oral)

**Comment:**

This paper proposed MIDI-DDSP, a structured hierarchical generative model which offers both detailed expressive controls (as in traditional synthesizers) as well as the realistic audio quality (as in black-box neural audio synthesis). Overall the reviews are very positive. All the reviewers unanimously agree that the paper is very well-written and presented a very convincing model and a meaningful step-up from the earlier work of DDSP. The authors also presented a well-documented website for the project and promised to release the source code. The reviewers raised some clarifying questions and minor corrections which the authors addressed during the response. Therefore, I vote for accept.